# The Combination of Low-Cost, Red–Green–Blue (RGB) Image Analysis and Machine Learning to Screen for Barley Plant Resistance to Net Blotch

**DOI:** 10.3390/plants13071039

**Published:** 2024-04-07

**Authors:** Fernanda Leiva, Rishap Dhakal, Kristiina Himanen, Rodomiro Ortiz, Aakash Chawade

**Affiliations:** 1Department of Plant Breeding, Swedish University of Agricultural Sciences (SLU), P.O. Box 190, SE-23422 Lomma, Sweden; fernanda.leiva@slu.se (F.L.); rodomiro.ortiz@slu.se (R.O.); 2Department of Plant and Agroecosystem Sciences, University of Wisconsin-Madison, 1575 Linden Dr, Madison, WI 53706, USA; 3National Plant Phenotyping Infrastructure, Helsinki Institute of Life Science, Biocenter Finland, University of Helsinki, Latokartanonkaari 7, 00790 Helsinki, Finland; kristiina.himanen@helsinki.fi

**Keywords:** barley, net blotch, disease symptoms, machine learning, RGB imaging

## Abstract

Challenges of climate change and growth population are exacerbated by noticeable environmental changes, which can increase the range of plant diseases, for instance, net blotch (NB), a foliar disease which significantly decreases barley (*Hordeum vulgare* L.) grain yield and quality. A resistant germplasm is usually identified through visual observation and the scoring of disease symptoms; however, this is subjective and time-consuming. Thus, automated, non-destructive, and low-cost disease-scoring approaches are highly relevant to barley breeding. This study presents a novel screening method for evaluating NB severity in barley. The proposed method uses an automated RGB imaging system, together with machine learning, to evaluate different symptoms and the severity of NB. The study was performed on three barley cultivars with distinct levels of resistance to NB (resistant, moderately resistant, and susceptible). The tested approach showed mean precision of 99% for various categories of NB severity (chlorotic, necrotic, and fungal lesions, along with leaf tip necrosis). The results demonstrate that the proposed method could be effective in assessing NB from barley leaves and specifying the level of NB severity; this type of information could be pivotal to precise selection for NB resistance in barley breeding.

## 1. Introduction

Climate change and steady population growth represent serious burdens to sustainable food production. The global human population has grown substantially over recent decades, which has noticeably strained the capacity of agricultural systems to provide a sufficient supply of food [1,2], to the point that many researchers are also shifting their attention to the valorization and use of wild species useful for human food [3,4,5]. At the same time, climate change—which has become increasingly noticeable—not only reduces crop yields, but also increases the risk that various plant pests and diseases will spread to new locations [6]. A primary goal of plant breeding programs is the identification of high-yield cultivars that can sufficiently meet consumers’ food demands [7], starting from the rediscovery of ancient varieties [8]. Barley is a major cereal crop that is mainly destined for direct human consumption but also transformed into processed food forms like bread, soups, stews, and health products [9]. These food products, including starch flour, flakes and pearled barley, are staple food in several countries, including Morocco, India, China, and Ethiopia (OECD, 2004). In Europe, up to 90% of barley is used for beer production (FAOSTAT, 2020); thus, this region is the largest producer and consumer of barley worldwide [10]. However, the dissemination and wider domains of plant pathogens represent a threat that can affect both barley grain quality and yields [7,11]. The fungal pathogen *Pyrenophora teres* is a major pest that affects barley [12].

*Pyrenophora teres* exists in two forms: the net form of net blotch (NB) caused by *P teres.* f. *teres* and the spot form of NB caused by *P teres.* f. *maculata* [13]. Both forms of this fungal pathogen will decrease grain yield and quality, ultimately contributing to noticeable economic losses. *P. teres* f. *teres* is a necrotrophic pathogen that grows at an intracellular level during the infection period [14]. Generally, the pathogen penetrates the epidermal layer within two days of infection [15]. Once inside the sub-stomatal chamber, the pathogen then creates secondary hyphae [16]. A prevalent technique for the laboratory evaluation of this pathogen involves detached leaf assays, in which lesion size is quantified, the number of spots is counted, and classification is based on lesion color. In whole-plant assays, inoculation is performed at a specified growth stage. Nevertheless, a comprehensive laboratory assessment needs to consider diverse parameters, with the most important encompassing the intricate interplay between the host, environment, and pathogen [8]. To mitigate the incidence and transmission of the disease, various management practices are implemented, e.g., the application of fungicides and seed treatments [17]. Additionally, strategies such as the utilization of biocontrol agents and crop rotation, which primarily target reductions in the primary inoculum, have been investigated for effectiveness in curbing infection [18]. Nevertheless, it has been shown that the best practical and long-term approach to controlling NB is the selection of resistant cultivars [19].

To date, resistance to the net form of NB has only been witnessed in a limited number of cultivars, while the majority of cultivars display only minimal susceptibility to the spot form of NB; this can greatly complicate the identification of resistant lines [13,19,20,21]. An additional challenge is that the evaluation of net blotch lesion severity traditionally involves visual assessments which employ a standardized 1–10 scale and are conducted as the disease progresses [22]. While this method has produced meaningful outcomes, the main drawbacks include a lack of precision and reproducibility, along with inherent time and labor intensiveness [23].

Over the last decades, efforts have focused on the development of high-throughput phenotyping methods to overcome these constraints [24,25,26]; one potentially robust approach relies on using image analysis to measure and analyze plant health [27,28,29,30]. This simple and low-cost solution mostly uses red–green–blue (RGB) imagery to extract information about the shape, texture, and color of plants, and has shown promise for the evaluation of different abiotic and biotic stresses even before the typical symptoms manifest [31,32]. Image acquisition can be performed through various methods, either manually [33] or via an automatic platform [34], and encompass various devices such as mobile phones [35], imaging chambers [36], high-throughput phenotyping facilities [37], or drones [38]. An example is the Phenocave, an automatic, low-cost, custom-built, and user-friendly system for image acquisition under indoor growth conditions [34]. This system has proven to be useful for capturing images with different cameras, e.g., RGB, multi-, and hyperspectral. Integrating these technologies with suitable analytical approaches can make it possible for researchers to differentiate between infected and healthy plants, as well as determine the severity, stage, and type of disease [39,40].

In the same sense, improvements in the software for image analysis have been accomplished in recent years; an example is ImageJ/Fiji [41,42], which is a platform dedicated to biological image analysis and contains plugins—written in Java—that can be used to tailor the image editing and analysis to a specific problem. Some of these plugins have been developed to provide machine learning approaches to analysis; for instance, the machine learning plugin “The Trainable Weka Segmentation” (TWS) [43] is a combination of image segmentation and machine learning algorithms that use the Waikato Environment for Knowledge Analysis. As such, almost any user can utilize this plugin, which does not require proficiency in coding. It has shown great promise in the classification of plants [44] as well as other domains within plant research [45,46,47].

Extensive research has specifically focused on using image analysis for the detection of NB. Notably, one prior study aimed to identify techniques that are relevant to the early detection of various diseases, including wheat yellow rust, stem rust, powdery mildew, potato late blight, and wild barley NB. Despite achieving 95% accuracy under field conditions by using deep learning approaches, the study primarily focused on early disease detection, along with the differentiation of various diseases; as such, the developed technique did not prove to be useful for following NB progression [48]. Another broad study that employed deep learning methods reported 75% accuracy for the identification of various spot and lesions; unfortunately, the applicability of this approach to specific cases, e.g., barley NB, remains unclear [39]. Therefore, while various techniques for identifying plant disease currently exist, most lack a simple and user-friendly interface that would allow large-scale application through the plant pathogen field.

The present study introduces a novel method for NB screening in barley which draws upon the Phenocave system for automatic image acquisition; more specifically, an RGB camera and the TWS plugin for image segmentation in ImageJ. The study was conducted in a growth chamber and involved three barley cultivars (resistant, moderately resistant, and susceptible).

## 2. Materials and Methods

### 2.1. Plant Material

The plant material used in this study comprised three spring barley cultivars selected on the basis of resistance to NB: Laureate (resistant); Firefoxx (moderately resistant); and Flair (susceptible). Approximately 60 seeds representing each cultivar were germinated on moist filter paper in petri dishes, prior to transplantation into soil. The seeds were germinated for three days under dark conditions at a temperature of 20 °C. The germinated seedlings were then transplanted to plastic pots (9 × 9 × 8 cm) filled with a mix of soil (Emmaljunga plantjord produced by Emmaljunga Torvmull AB) containing osmocote, a long-acting fertilizer, and oxywet, a product that helps retain soil moisture; three seedlings were planted in each pot. Thus, each cultivar had a total of 20 pots; all 60 pots were placed on three trolleys in a completely randomized order provided by the design.crd() function in the agricolae package [49] for R software version 4.2.

The seedlings were left to grow for 10 days, after which the full emergence of a second leaf was observed in all plants (n = 60). This step, along with the subsequent parts of this study, were conducted in controlled conditions in a climate chamber at the Biotron research facility, which is located at the Swedish University of Agricultural Sciences in Alnarp, Skåne, southern Sweden. The plants were grown at a light intensity of 250 μmol m^−2^ s^−1^ with a photoperiod of 12 h/12 h day/night, a relative humidity of 65%, and a constant temperature of 19 °C.

### 2.2. Inoculation Protocol

The isolated fungal pathogen *Pyrenophora teres* f. *teres* was retrieved from glycerol stocks stored at −80 °C and plated onto Petri dishes with 20% V8 media. The plates were incubated for three days at 19 °C and a 12/12 h white light/dark cycle, after which the plates were moved to a UV chamber for a five-day 12/12 h UV/darkness regimen (constant temperature of 19 °C). This was followed by an 8/16 h UV/darkness regimen for another 10 days.

After incubation, the plates were flooded with approximately 10 mL of sterile water and the contents were scraped into a beaker. The initial spore count of 55,000 spores/mL in the resulting inoculum was adjusted to 27,500 spores/mL using a hemocytometer. Finally, 1 µL of the surfactant Tween 20 was added to the suspension to facilitate spore adhesion to the leaf surface.

A subset of the 60 pots was distributed among 12 trays according to a two-factorial design including all three of the cultivars, two treatments, and two replicates (Figure 1A). The design was generated using the design.ab() function in the agricolae R package. A total of four pots representing each cultivar were placed into a tray. Next, the second leaves from plants were fixed to a horizontal board using rubber bands; this kept the leaves flat against a white background (Figure 1B). Each tray then consisted of two clusters of 5–6 leaves. The two treatments included inoculation, under which each leaf was brushed with 30 μL of inoculum, and control, under which each leaf was brushed with a mixture of sterile water with 0.02% Tween. A control treatment was included to record data concerning stress symptoms that were unrelated to net blotch infection. Immediately following infection, the trays were covered with black plastic bags for 24 h to maintain high humidity and darkness.

### 2.3. Image Acquisition

Top-down images were captured on a daily basis using a Canon EOS 1300D camera (Canon, Tokyo, Japan), with a Canon EF-S 50 mm f/1.8 STM lens, that was mounted onto the Phenocave automated imaging gantry system [34]. The camera parameters used when capturing images were a shutter speed of 1/50 s, an aperture stop of f/16, and ISO 200. The images were saved in the JPEG file format. The capturing of images started one day after infection (DAI) and proceeded until 15 DAI. The Phenocave was set up in a way that each cluster of leaves could be photographed separately; this resulted in two pictures per tray at each time point. An example of an image taken from two clusters of leaves is shown in Figure 2.

### 2.4. Image Processing

The resulting 720 images were processed using Raw Therapee v5.8 to crop the images to a size of 900 × 1024 pixels. The free software ImageJ/Fiji (ver.1.53c.) [41], with the Trainable Weka Segmentation (TWS) plugin (ver.3.2.35) [43], was used to segment the images. When training the model in TWS, one cluster per tray was randomly selected, using only images collected at 1, 7, and 15 DAI.

Each image of the training set was opened in the TWS interface, after which leaf regions were manually marked according to the following six classes: (1) healthy leaf area; (2) necrotic lesion area; (3) chlorotic lesion area; (4) fungal lesion area; (5) leaf tip necrosis; and (6) background. The classes were selected according to the fungal symptoms given for classes 2 to 4, and if necrosis was observed at the leaf tip (class 5). This type of necrosis was a common form of stress that appeared in both infected and control leaves. The model was built using the Random Forest algorithm with the training features Gaussian Blur, Sobel filter, Hessian, Difference of Gaussians, Membrane projections, Variance, Mean, Minimum, Maximum, Median, Anisotropic Diffusion, Bilateral, Lipschitz, Kuwahara, Gabor, Entropy, and Neighbors. The membrane thickness was set to ‘1’, with a patch size of 19. Minimum and maximum sigma were set to 1 and 16, respectively. Using images from 50% of the total infected and control plants yielded a dataset with 215,913 rows of data and 231 features, with each row containing features from a manually-selected area of five pixels.

The resulting model was able to classify the entire image, i.e., provide an image that is segmented into the six categories described above. These images were then used to identify any wrongly classified regions in the image, which were manually corrected to improve model performance. 

### 2.5. Model Performance

The training set data were further analyzed in R to investigate the class-wise accuracy and variable importance of the potential prediction model. The R package caret version 6.0 [50] was used to conduct a 10-fold cross-validation, which was repeated five times. Class-wise classification accuracy was extracted from each iteration by computing a confusion matrix for the assigned and predicted classes for each pixel. The final model produced in the R package caret was used to assess variable importance. Finally, the trained random forest model was applied to the remaining images collected in the experiment.

## 3. Results

In this study, a total of 60 plants were inoculated with the fungal pathogen that causes NB. Visual scoring and RGB imaging of the leaf clusters was already performed one DAI. The RGB images were processed using the TWS plugin of Fiji/ImageJ software version 2317, during which pixels were classified into five different categories: healthy; chlorotic; necrotic; fungal; and leaf tip necrosis. A total of 65 leaves across six trays were infected. The total number of inoculated leaves across the three barley cultivars, including the percentage of leaves that developed infection symptoms, are detailed in Table 1.

The results obtained from the training and test sets were cross-validated to verify the accuracy of classification. The results of this cross-validation analysis revealed high classification accuracy for all classes of 0.99 (Table 2), with slightly differences in decimals of approximately 0.004 for each class evaluated. However, 0.78% of the total pixels of images were wrongly classified as belonging to different categories. For example, in the case of pixels representing the background, 0.05% of these pixels were misclassified as leaf tip necrosis. Moreover, 0.11% of the pixels representing a healthy leaf were misclassified as fungal lesions, while the same proportion (0.11%) of pixels representing chlorotic lesions were misclassified as necrotic lesions. The proportion of misclassified pixels increased in the case of fungal lesions, with 0.31% of the pixels misclassified as necrotic lesions. About 0.2% of pixels representing leaf tip necrosis were misclassified as necrosis. Although these cases of misclassification provide evidence that the process was not entirely accurate, the fact that less than 1% of the analyzed pixels were misclassified can be considered as an insignificant error; in this way, we believe that the presented image analysis approach can provide reliable and accurate results in a timely manner. The final model, which was trained using the full training dataset, was applied to all of the images, including the training set images and images left out of the training set. An example of how the model performed on images not included in the training set can be seen in Figure 3, with pixels in two infected clusters of leaves at 15 DAI classified into one of the six classes.

## 4. Feature Importance

A total of 20 features, such as entropy corresponding to texture information and hue and saturation corresponding to color, were used when creating the model for classification. Moreover, features related to noise reduction, such as Kuwahara, Bilateral, and Anisotropic diffusion reduction, were included. All of these features were then included in the analysis of variable importance to determine which features provide the most information about visible NB symptoms. The results, illustrated in Figure 4, revealed that the features related to color are highly important, with hue receiving a score of 100% for importance and saturation receiving a score of 24% for importance. Features related to noise reduction were also important for the model, with Kuwahara receiving a score of 40%, while the feature “entropy” also received a high score (37.5%) for variable importance.

To capture the progression of the disease in barley plants, the pixel counts describing affected areas of leaves were plotted across time. As can be observed in Figure 5, the period from the 21st of February (1 DAI) to the 7th of March (15 DAI) showed significant changes in the health of leaves inoculated with *Pyrenophora teres* f. *teres*. The infected plants, denoted with the letter A, showed more significant changes when compared to control plants, denoted with the letter B. Although all of the plants belonging to the same cultivar showed similar trends in the experiments, there were several cases in which plants from the same cultivar had significant differences in NB progression. For example, plants representing the cultivar Flair in replicate I were highly affected by NB, with symptoms of fungal lesions, while the Flair plants in replicate II were classified as resistant. Moreover, plants belonging to the cultivar Laureate in replicate I showed some damage caused by the infection, such as fungal lesions, while the Laureate barley plants in replicate II were found to be the most susceptible to NB, with over 75% of the leaf area showing symptoms, especially fungal lesions. Concerning the cultivar Firefoxx, the results confirmed that this cultivar is highly resistant to NB. In replicate I, leaf tip necrosis was observed in less than 25% of the leaf area, with 30% of the leaf area affected by chlorotic lesions. A similar trend was observed in replicate II, with one group of plants showing leaf tip necrosis in less than 40% of the leaf area and chlorotic lesions affecting approximately 30% of the leaf area. As such, plants representing the Firefoxx cultivar demonstrated very mild symptoms across both treatments and replicates.

## 5. Discussion

This study evaluated the potential of a novel approach in the detection and quantification of the symptoms and severity of NB in barley which is an economically important disease in the temperate region. NB in barley also has a high broad sense heritability of 0.8 to 0.85, as documented previously [51], making it a suitable model system to evaluate imaging for disease detection. The results indicated that using a machine learning approach can provide utility in detecting differences among various cultivars. In addition, the present study evaluated two affordable, user-friendly, and time-efficient approaches. The low-cost image acquisition system Phenocave [34] was developed for indoor growth conditions and takes images using a regular RGB camera and two other imaging sensors (multi- and hyperspectral). We only analyzed RGB images, while the use of spectral data for disease screening can be explored in future studies. The performance of the TWS [43] plugin of the free software Fiji/ImageJ in image segmentation with different machine-learning algorithms was also assessed.

The features used to build the classification model were included in a variable importance analysis, the results of which revealed the most relevant characteristics to evaluate during RGB image processing. The analysis identified hue as the most significant feature (100% importance), followed by noise reduction via the Kuwahara filter and texture “entropy” (Figure 4). Although all of the other features contributed to the model performance, a second model could be created using a feature importance threshold of 25%. While the combination of all of the features provided a classification accuracy of 99%, it could be expected that a similar accuracy could be obtained by only including the features with the largest contributions to performance, as this would translate to large improvements in computational speed. The class-wise prediction analysis results revealed a certain degree of misclassifications; this may be explained by the sheer number of features used to create the model or by the number of samples analyzed. In this study, we only investigated three time points to evaluate whether the developed model is efficient, affordable, and easy-to-use. It is important to note that the inclusion of a longer study period of disease progression and obtaining more images, i.e., a larger sample size, could decrease the number of misclassifications. Nevertheless, only 0.78% of the classifications made based on samples collected from 60 plants over 15 days were wrong, which is very low, and essentially negligible; this provides strong evidence of the value of the presented approach.

Image data were systematically recorded over a span of 15 days that encompassed the period from 1 to 15 DAI. This experimental design enabled the detailed tracking of NB disease progression. The predictive capacity of the method not only facilitated the discrimination of resistant and susceptible cultivars, but also provided insights into the temporal dynamics of disease evolution. Notably, the susceptible cultivar Flair exhibited the highest percentage of leaves with signs of infection, whereas the moderately-resistant cultivar Firefoxx displayed the lowest percentage of infection symptoms over the analyzed leaf area, which even fell below the rates observed for the resistant Laureate cultivar. Although the three cultivars were chosen based on susceptibility to NB, the experiments provided a highly nuanced understanding of how the barley cultivars respond to fungal infection, e.g., a moderately-susceptible cultivar demonstrated better performance than a susceptible cultivar following inoculation. The described methodology, along with the quantitative results, provide strong evidence for the efficiency of combining machine learning with RGB image analysis for screening NB in barley. Furthermore, as the current standard for detecting NB in barley—manual observations—is time- and labor-intensive, the presented approach could be widely applicable to both research and agricultural contexts due to its efficiency.

## 6. Conclusions

The current study shows the potential viability of an approach which combines machine learning with RGB image analysis for screening resistance to NB. The cost-effective Phenocave system enabled rapid image acquisition, and proved to be highly advantageous in terms of time savings. Furthermore, the application of the user-friendly TWS plugin, which provided access to machine learning approaches, translated to the successful detection of various symptoms attributable to NB. Even though some misclassifications occurred, the share of these mistakes to the overall correct classifications across all of the evaluated categories was minimal (0.78%) when compared to the 99% accuracy for different symptoms of NB. Additionally, certain features, such as entropy, noise reduction filters like Kuwahara, and specific color values of hue and saturation, were found to be highly accurate at classifying pixels into the correct category of disease-specific symptoms, and should be examined further to verify the presented findings. This study not only contributes valuable insights into NB detection in barley, but also demonstrates the potential application of the presented method in identifying other diseases with similar characteristics across various plants. For instance, the quantitative differentiation of disease symptoms among diverse barley cultivars suggests broader implications for the field of plant pathology. In light of ever-increasing climate change, the further exploration of the identified features could yield various methodologies that can be used to detect and follow the progression of multiple plant disease in a highly efficient manner. Also, in order to advance our understanding of environmental monitoring net blotch disease progression, as a future study, there is a promising opportunity to adapt this indoor condition screening method for real outdoor applications with different tunings.

## Figures and Tables

**Figure 1 plants-13-01039-f001:**
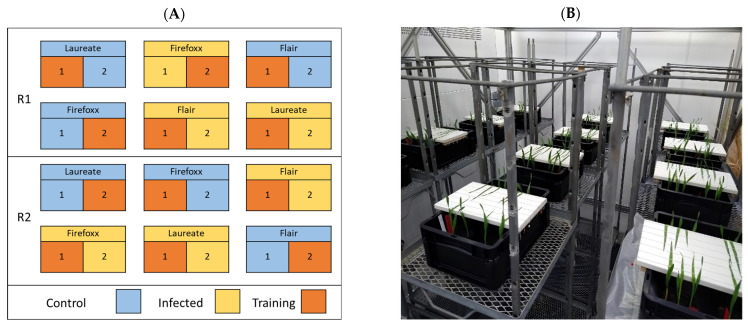
(**A**) Distribution of cultivars under the two-factorial experimental design, which included two replicates (R1 and R2) and two treatments (control (blue) and infected (yellow)). The numbers 1 and 2 represent plants that represent the same cultivar but received different treatments, respectively. The plants marked with an orange color were used to build the training set for the applied model; (**B**) Image of the 12 plant trays in the chamber.

**Figure 2 plants-13-01039-f002:**
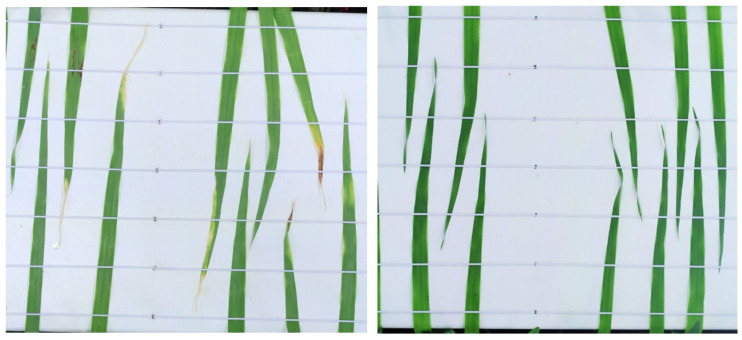
Example of an image showing two clusters of leaves; the left photo shows the disease symptoms on the leaves, while the right photo shows sound leaves that were not infected.

**Figure 3 plants-13-01039-f003:**
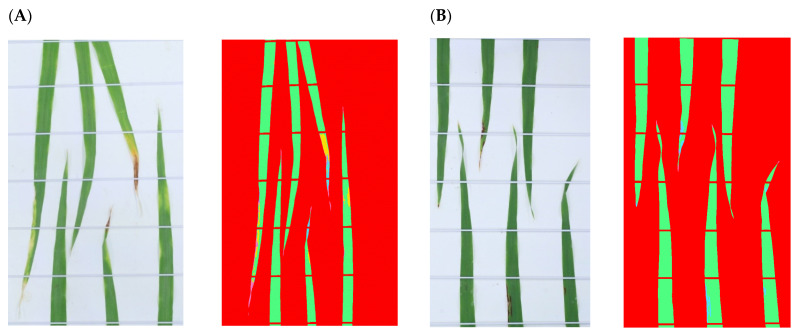
An example of the application of the final model to test set images taken 15 DAI. (**A**) Susceptible cultivar, original (left) and processed (right); (**B**) resistant cultivar, original (left) and processed (right). The various classes are shown using the following colors: background (red); healthy leaf area (green); chlorotic lesion (yellow); necrotic lesion (purple); fungal lesion (blue); and leaf tip necrosis (magenta).

**Figure 4 plants-13-01039-f004:**
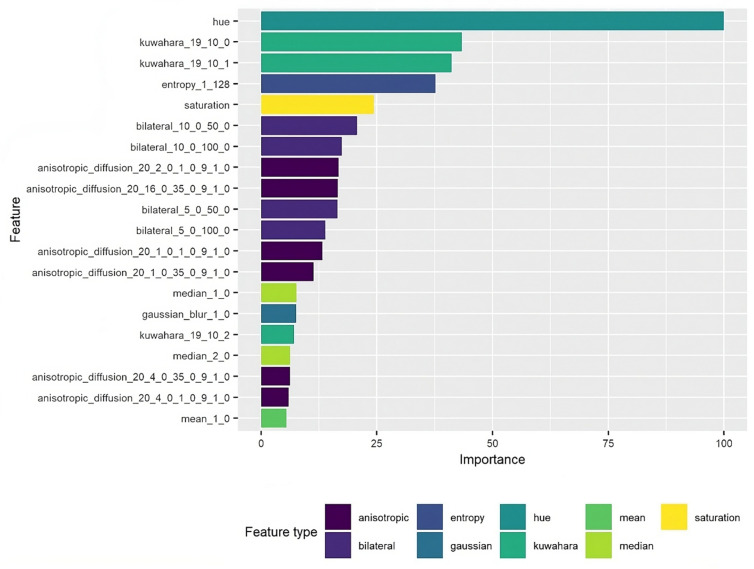
Variable importance plot, with variables towards the top contributing more to predictive power than variables toward the bottom. Variable importance is shown as a percentage. The features were categorized by type and identified by color in the feature type chart.

**Figure 5 plants-13-01039-f005:**
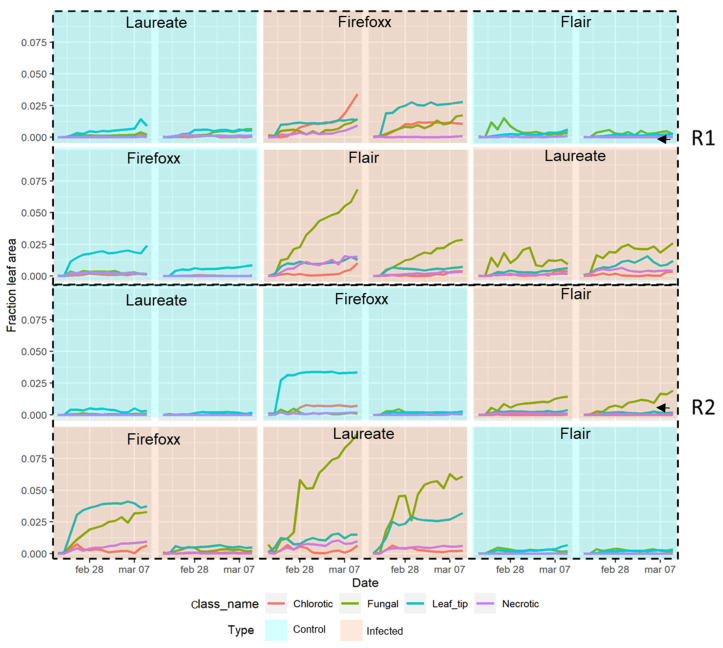
Disease progression at 0, 7, and 15 DAI in control (cyan) and inoculated (red) sets of plants. The R1 and R2 correspond to the number of replicates. The cultivar is specified at the top of each plot, while the colored lines indicate different classes of disease severity.

**Table 1 plants-13-01039-t001:** Total number of inoculated leaves for the three tested barley cultivars, including the percentage of leaves that developed infection symptoms.

Cultivar	No. Inoculated Leaves	Percentage Infected
Firefoxx	23	43.5%
Flair	23	69.6%
Laureate	19	52.6%

**Table 2 plants-13-01039-t002:** Class-wise confusion matrix of prediction accuracy and variable importance. TBA: statistics concerning spotty disease infection.

		Reference					
		Background	Healthy	Necrotic	Chlorotic	Fungal	Leaf tip necrosis
Prediction	Background	3,797,616	125	19	0	21	159
	Healthy	468	1,988,363	37	59	318	55
	Necrotic	27	870	91,082	73	958	470
	Chlorotic	36	225	36	66,011	40	162
	Fungal	251	2319	217	45	300,801	147
	Leaf tip necrosis	1971	400	16	29	184	223,780
Overall accuracy		0.999	0.998	0.996	0.997	0.995	0.996

## Data Availability

The original contributions presented in the study are included in the article, further inquiries can be directed to the corresponding author.

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
