# Peer review of "The Combination of Low-Cost, Red–Green–Blue (RGB) Image Analysis and Machine Learning to Screen for Barley Plant Resistance to Net Blotch"

_plants, 2024, doi:10.3390/plants13071039_

Round 1

Reviewer 1 Report

Comments and Suggestions for Authors

The authors propose a manuscript titled “The combination of low-cost, red-green-blue (RGB) image analysis and machine learning to screen for barley plant resistance to net blotch”. The article give original information and is well written. The study were conducted on on the increasing plant diseases as net blotch (NB), a foliar disease which significantly decreases barley grain yield and quality. The study presents a novel screening method for evaluating NB severity in barley trought an automated RGB imaging system, together with machine learning, in order to evaluate different symptoms and the severity of NB on three barley cultivars with distinct levels of resistance to NB (resistant, moderately resistant, and susceptible). The results demonstrate that the proposed method could be effective in assessing NB from barley leaves and specifying the level of NB severity. The manuscript deserve of few other crucial informations after which will be able published.

Abstract

·         Lines 13-14. Move in the introduction this period: “Climate change and steady population growth represent serious burdens to sustainable food production”.

Introduction

Few remarks. Add some references as follow suggested (in bold) because sometimes the concept are already exisisting:

·         Lines 31-33. The global human population has grown substantially over recent decades, which has noticeably strained the capacity of agricultural systems to provide a sufficient supply of food [choose reference], to the point that many researchers are also shifting their attention to the valorization and use of wild species useful for human food [e.g. Ben Mahmoud et al. 2024 choose other 2 references];

·         Lines 35-36. A primary goal of plant breeding programs is the identification of high-yield cultivars that can sufficiently meet consumers' food demands, also starting from the rediscovery of ancient varieties [Abenavoli et al. 2020];

·         Lines 70-71. Over the last decades, efforts have focused on the development of high-throughput phenotyping methods to overcome these constraints [choose reference];

·         Lines 75-78. Image acquisition can be performed through various methods either manually or via an automatic platform, and encompass various devices such as mobile phones, imaging chambers, high-throughput phenotyping facilities, or drones. I don't understand this comment! Please specify better which method was used in your work, and in any case consider at least a references for each method here indicated.

·         Lines 78-106. Summarize this period of the introduction.

Reference to be added

·         Ben Mahmoud, K.; Mezzapesa, G.N.; Abdelkefi, F.; Perrino, E.V. Beta macrocarpa Guss. in Tunisia: nutritional and functional properties of the underutilized wild beet in relation to soil characteristics. Euro-Mediterranean Journal for Environmental Integration 2024. In press

·         Perrino, E.V.; Mahmoud, Z.N.A.; Valerio, F.; Tomaselli, V.; Wagensommer R.P.; Trani, A. (2023. Synecology of Lagoecia cuminoides L. in Italy and evaluation of functional compounds presence in its water or hydroalcoholic extracts. Scientific Reports 2023. 13: 20906.  doi: 10.1038/s41598-023-48065-w

·         Abenavoli, L.; Milanovic, M.; Procopio, A.C.; Spampinato, G. et al. Ancient wheats: beneficial effects on insulin resistance. Minerva Medica 2020. 112, 641-650. Doi: 10.23736/S0026-4806.20.06873-1

2. Materials and Methods

Only one observation.

·         Please give the accession of the plant material used in the study and specifically the three spring barley cultivars selected.

3. & 4. Results and Discussion

Few observations.

·         Line 215. Please specify what is meant 0.995 to 0.999;

·         Fig. 4 is not clear. Please increase the resolution.

4. Conclusions

In the conclusion I suggest to writing two words on the aspect concerning the governance and the perspectives on future studies.

References

Please give Doi when is available

Author Response

Thank you for the suggestions and for pointing out some failures that are crucial for a better article.

Please find all the answers to the corresponding questions and comments that were given for consideration.

Reviewer 1.

Introduction

Lines 13-14. Move in the introduction this period: “Climate change and steady population growth represent serious burdens to sustainable food production”.

  • This line was moved and the text was also modified to correspond to this change.

Few remarks. Add some references as follow suggested (in bold) because sometimes the concept are already exisisting:

  • Lines 31-33. The global human population has grown substantially over recent decades, which has noticeably strained the capacity of agricultural systems to provide a sufficient supply of food [choose reference], to the point that many researchers are also shifting their attention to the valorization and use of wild species useful for human food [e.g. Ben Mahmoud et al. 2024 choose other 2 references];
  • Lines 35-36. A primary goal of plant breeding programs is the identification of high-yield cultivars that can sufficiently meet consumers' food demands, also starting from the rediscovery of ancient varieties [Abenavoli et al. 2020];
  • Lines 70-71. Over the last decades, efforts have focused on the development of high-throughput phenotyping methods to overcome these constraints [choose reference];
  • It has been added the corresponding listed references, including new ones on the respected marked sections.

Lines 75-78. Image acquisition can be performed through various methods either manually or via an automatic platform, and encompass various devices such as mobile phones, imaging chambers, high-throughput phenotyping facilities, or drones. I don't understand this comment! Please specify better which method was used in your work, and in any case consider at least a references for each method here indicated.

  • It has been added corresponding references to support the lines, which specify the different tools that can be used in digital phenotyping. However, as this is just part of the introduction, we prefer to mention some lines after that is well explained the methods used in this article and the use of the Phenocave platform in next line.

Lines 78-106. Summarize this period of the introduction.

  • There is a summary of the introduction that encompass the idea of the work and the combination of the different sections explained in the introduction.
  1. Materials and Methods

Please give the accession of the plant material used in the study and specifically the three spring barley cultivars selected.

The three spring barley cultivars are Laureate (resistant); Firefoxx (moderately resistant); and Flair (susceptible). These are market cultivars and thus there is no accession number for them.

Results and Discussion

Line 215. Please specify what is meant 0.995 to 0.999

  • This line has been modified for a better understanding on this values and the differences that we wanted to point out.

Fig. 4 is not clear. Please increase the resolution.

  • This image has a good quality, but it was lost during the change from word to PDf we will make sure to upload a better version.

Conclusions

In the conclusion I suggest to writing two words on the aspect concerning the governance and the perspectives on future studies.

  • Future studies and perspectives was added in lines 487-489.

Reviewer 2 Report

Comments and Suggestions for Authors

Dear Authors. Overall, I rate the manuscript as an interesting paper whose subject matter is in line with current research trends. However, I have some comments that I give to the Authors for consideration. I have placed all my comments directly in the attached pdf file. In particular, please note the comment on line 288.

Author Response

Thankyou for your valueable feedback. We tried to answer them to the best of our ability. 

Introduction

It is worth pointing out that there is also feed barley

  • It has been added a new line and reference pointing out this as well as mention its importance in other countries than Europe.

Necrotic and necrotrophic. These terms are not interchangeable and mean different things.

  • We truly apologize for this confusion of terms. It has been modified and verified that there are not more errors like this within the text.

A brief information on the heritability of barley resistance to P. teres, supported by an appropriate reference.

  • We have now included the heritability in the beginning of the discussion section.

Firefox or Firefoxx?

  • In effect, there was a misspelling with this word. Indeed, it is Firefoxx. It has been verified that there was only this misspell error in this line.

Why did the Authors use JPEG images and not a lossless compression format such as BMP or TIFF? A lossy compression always results in lower image quality.

  • In this case, several factors were considered. It is true that JPGE lower image quality but each format has its advantages and disadvantages. For instance, file size: JPGEs are typically much smaller, compared to TIFF files. Although this can result in some loss of image quality, the compression is often optimized to balance file size and visual quality. In addition, for the imaging sampling and processing considering a low-cost application we chose this format as well for the speed on the image processing.

Figure 2, Please check the figure caption carefully.

  • This caption has been modified, there was a mix with the legend and the position of the images.

Improve image quality Figure 4 and 5.

  • During the compression and changing the format from word office to PDF, the images lost the quality. The original images are in good quality and we will ensure to conserve that in the new updated file.

The most important comments.

Do the Authors use the term "RGB images" to describe true-colour images (24-bit) or 32-bit images?

  • Yes, the general term for RGB images and also in this case is referring to true-color images (24-bits)

How was the conversion from RGB to HS... (what?) done? Do the Authors have in mind the HSI, HSV or HSB model? The results are missing a reference to the third color descriptor. However, color is a multidimensional trait, and limiting its characteristics to two descriptors is not quite correct. Something should be done about this in the revised version of MS.

  • In this case, using the Weka classifier plugin for Fiji/ImageJ, there is an internal extraction of the HSB features. It does not inherently convert images to the HSB (Hue, Saturation, and Brightness) color space for processing. Instead, it operates on the pixel values directly, irrespective of the color space.

Why was the L, a, b model not used? I understand that the HS(...) model is more observer-friendly and more intuitive, but the color description in the L, a, b space may be better for this type of research.

  • The use of Lab model is correct. It provides better results and more information. However, as it was previously described. In order to use the plugin Weka classifier, the images are considered irrespective of the color space.